# PromptCoT: Align Prompt Distribution via Adapted Chain-of-Thought

## Abstract

Diffusion-based generative models have exhibited remarkable capability in the production of high-fidelity visual content such as images and videos. However, their performance is significantly contingent upon the quality of textual inputs, commonly referred to as "prompts". The process of traditional prompt engineering, while effective, necessitates empirical expertise and poses challenges for inexperienced users. In this paper, we introduce PromptCoT, an innovative enhancer that autonomously refines prompts for users. The design of PromptCoT is based on the observation that, prompts resembling textual information corresponding to high-quality images within the training set tend to yield superior generation performance. As such, we fine-tune the pre-trained Large Language Models (LLM) using a curated text dataset comprising solely of high-quality visual content descriptions. By doing so, the LLM becomes capable of capturing the distribution of high-quality training texts, enabling it to generate aligned continuations and revisions to boost the original texts. Nonetheless, one drawback of pre-trained LLMs is their tendency to generate extraneous or irrelevant information. To enhance the alignment between the original text prompts and the refined counterparts, we leverage the Chain-of-Thought (CoT) mechanism. CoT can extract and amalgamate crucial information from the aligned continuation and revision, enabling reasonable inferences based on the contextual cues to produce a more comprehensive and nuanced final output. Considering computational efficiency, instead of allocating a dedicated LLM for prompt enhancement to each individual model or dataset, we integrate adapters that facilitate dataset-specific adaptation, leveraging a shared pre-trained LLM as the foundation for this process. By fine-tuning these adapters independently, we can adapt PromptCoT to new datasets with minimal increase in training cost and memory usage. We assess the performance of PromptCoT on widely-used latent diffusion models for image and video generation to validate the effectiveness. The results demonstrate significant improvements in key performance metrics.

## 1 Introduction

In recent years, deep generative models have made notable advancements, specifically with the introduction of diffusion probabilistic models (DPMs). These models have exhibited exceptional capabilities in generating a wide range of visually compelling and high-fidelity visual contents, such as images and videos, as evidenced by notable contributions in the literature Song & Ermon (2019); Ho et al. (2020); Song et al. (2020b;a); Dhariwal & Nichol (2021); Ramesh et al. (2022); Saharia et al. (2022); Rombach et al. (2021).

By harnessing textual inputs as conditional guidance, diffusion models have the ability to generate visual outputs that align with the corresponding input text, utilizing an iterative denoising procedure. This technological advancement has paved the way for revolutionary applications, including notable examples such as DALL-E 2 Ramesh et al. (2022), Stable Diffusion Rombach et al. (2021), MagicVideo Zhou et al. (2022a), among others.

Nevertheless, the quality of the generated content is intricately tied to the caliber of the textual prompts provided to the generative model. Human inputs tend to be informal and straightforward, which may impede the expression of the desired scene with the desired level of depth. Additionally, the text encoder within the generative model may not fully comprehend the semantic nuances present in the human-generated text, resulting in notable disparities between the encoded textual guidance and the user's intended meaning. Diffusion probabilistic models (DPMs) are commonly trained

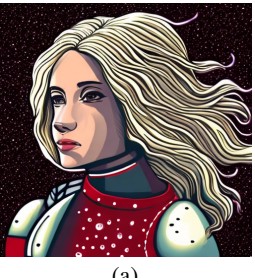 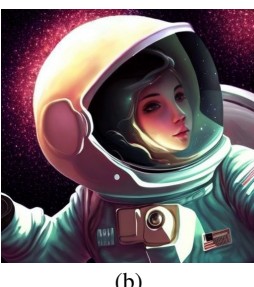 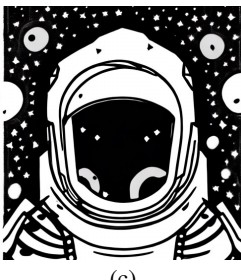 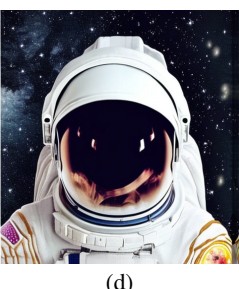

(a)                    (b)                    (c)                    (d)

Figure 1: **Impacts of PromptCoT.** (a) and (c) show the images generated with the original text prompts, and (b) and (d) show the images generated with the text prompts refined by PromptCoT. The text prompt for (a), (b), (c) and (d) are: 1) "highly detailed portrait of a hopeful pretty astronaut lady with a wavy blonde hair, by Jamini Roy , 4k resolution, nier:automata inspired, bravely default inspired, vibrant but dreary but uplifting red, black and white color scheme!!! ((Space nebula background))" ; 2) "Astronaut portrait of Silica from the game Bravely Default II by Jamini Roy", and 3) "highly detailed portrait of a hopeful pretty astronaut lady with a wavy blonde hair, by Pablo Picasso, 4k resolution, nier:automata inspired, bravely default inspired, vibrant but dreary but uplifting red, black and white color scheme!!! ((Space nebula background))",and 4)"Portrait Of A Beautiful Astronaut Girl Canvas Art Print" respectively.

on extensive text-vision pairs acquired through web-scraping techniques Schuhmann et al. (2022). Our observation reveals that the distribution of the text dataset might not be congruent with the linguistic style employed by layman users. Furthermore, even in cases where the training text data aligns with the desired style, the quality can exhibit substantial variations due to the presence of meaningless words or extraneous information within the text data. This intricacy further complicates the establishment of a clear and unambiguous mapping between the text and the corresponding image.

As a result, there is an immediate imperative to develop a methodology that can effectively align prompts, consequently augmenting the image generation performance in generative models. Although data cleaning and model fine-tuning have been considered potential solutions, these methods often entail drawbacks such as high costs, instability, and time intensiveness. Another alternative is manual prompt engineering, which involves refining prompts to optimize generation performance. However, this empirical task traditionally demands the expertise of experienced professionals, thereby posing a significant challenge for individuals lacking relevant experience.

In our study, we observe a noticeable trend that prompts, which resemble those found in the training set, usually lead to superior generative performance. Stemming from this observation, we propose PromptCoT, a novel prompt booster that leverages the power of pre-trained Large Language Models (LLMs) and incorporates the Chain-of-Thought (CoT) mechanism to learn high-quality prompt expressions from the training texts of generative models. Specifically, we carry out the fine-tuning of LLaMA Touvron et al. (2023), a widely-used pre-trained Large Language Model, on two distinct datasets we've prepared. With a text-continuation dataset that appends aligned details to original prompts, and a text-revision dataset that rewrites original prompts to aligned prompts, we enable LLaMA to refine prompts that better match the distribution of the text data used for training the diffusion models. To further enhance the performance of LLMs by combining the advantages of both text-continuation and text-revision, we construct a dataset using the CoT mechanism assisted by ChatGPT. This CoT dataset is designed to enable LLMs to reason and generate text that follows a logical and coherent flow. By fine-tuning LLMs on this CoT dataset, we can enhance their reasoning ability and augments their capacity to generate high-quality text that is both contextually relevant and logically coherent.

To accommodate the varying training sets of different generative models, we incorporate a parameter-efficient adaptation design into the training pipeline of PromptCoT, augmenting a pre-trained base booster with specific lightweight adapters that are capable of aligning text distributions for various generative models across multiple tasks. We demonstrate the effectiveness of PromptCoT through extensive experiments on widely-used latent diffusion models for image and video generation, showing significant improvements in key performance metrics such as Fréchet Inception Distance, aesthetic score, and CLIP-similarity.

Our main contributions are:

• We propose PromptCoT, an innovative prompt refiner that aligns input prompts with the text distribution employed during the training of diffusion models. By accomplishing this alignment, PromptCoT effectively activates generative models and enhances their performance.

• We explore a new optimization scheme for improving prompt quality by leveraging the power of pre-trained LLMs and CoT mechanisms. And we construct datasets to facilitate the learning of high-quality prompt distribution from the training texts of generative models.

• We demonstrate that allocating a dedicated Large Language Model (LLM) for each diffusion model is not a requirement. Instead, we propose an innovative scheme where a set of lightweight adapter weights suffices for each dedicated diffusion model. These adapters can share a shared base pre-trained LLM, resulting in a considerable reduction in memory footprint.

• We show the effectiveness of PromptCoT through extensive experiments on widely-used latent diffusion models for image and video generation, showing significant improvements in key performance metrics.

## 2 RELATED WORK

### 2.1 TEXT-TO-IMAGE GENERATIVE MODELS

Text-to-Image Generative Models operate by taking natural language descriptions as input and generating corresponding images as output. One of the recent popular model is DALL·E 2 Ramesh et al. (2021). It utilize CLIP Radford et al. (2021) to align the text and image embeddings. By conditioning the diffusion probabilistic generator on the textual embedding, DALL·E 2 is able to produce photorealistic images that correspond to the given textual description. Later, Google's Imagen Saharia et al. (2022) and Parti Yu et al. (2022) were proposed by gradually simulating the spread of noise into the original image to reveal the desired image. Specifically, both Parti and Imagen combine autoregressive and diffusion. The application of diffusion probabilistic models has also been extended to the domain of video generation. The Video Diffusion Model Ho et al. (2022), built upon the foundations of diffusion models, enables the sequential generation of high-quality video frames. To address the substantial computational requirements associated with video generation, MagicVideo Zhou et al. (2023) was introduced, combining latent diffusion and attention models. MagicVideo utilizes a frame-wise lightweight adapter and an attention module to effectively adjust the image-to-video distribution and capture temporal dependencies across frames.

### 2.2 LARGE LANGUAGE MODELS

Large Language Models (LLMs) are powerful deep learning models for various natural language processing tasks. The most popular LLMs are the GPT Radford et al. (2019); Brown et al. (2020b) series models developed by OpenAI, which are based on the decoder component of the transformer architecture. Another LLM is Meta's OPT Zhang et al. (2022), which is open-sourced and performs similarly in performance to GPT-3. However, GPT-3's massive size of 175B parameters requires significant computing power and resources, which makes it challenging for researchers to explore. In contrast, LLaMA Touvron et al. (2023); von Werra et al. (2023), StableLM Andonian et al. (2021), as well as the instruction-following Alpaca model Taori et al. (2023) are smaller and more performant, achieve comparable results to ChatGPT with far fewer parameters (7B). For specific tasks like conversational applications, ChatGLM Zeng et al. (2022); Du et al. (2022) can generate coherent and contextually relevant responses in dialogue systems.

### 2.3 PARAMETER-EFFICIENT FINE-TUNING

The goal of parameter-efficient fine-tuning is to attain comparable performance to fine-tuning on a specific downstream task while using the fewest trainable parameters possible. According to Aghajanyan et al. (2020), common pre-trained models generally have a very low intrinsic dimension, and LoRA Hu et al. (2021) learns low-rank parameterizations to enhance tuning efficiency based on that. Except reducing the number of parameters needed for fine-tuning, other approaches try to attach pre-trained parameters to reduce training time. Adapter training Houlsby et al. (2019); Pfeiffer et al. (2020) utilizes dynamic pre-trained adapters for different tasks and languages to reduce adaptation time. Compacter Mahabadi et al. (2021) combines both concepts and builds on top of adapters, low-rank optimization, and parameterized hypercomplex multiplication layers.

### 2.4 PROMPT ENGINEERING

Prompt Engineering is to optimize the outputs of language models with specific input prompts Brown et al. (2020a); Schick et al. (2020); Liu et al. (2021); Ding et al. (2022). Discrete text prompts Hu et al. (2017) serve as starting points for the model's language generation, and are used to generate responses in dialogue systems. Beyond discrete prompts, Lester et al. (2021); Wei et al. (2021) explores prompt tuning to learn soft prompts to perform specific downstream tasks, which provide more context-aware guidance to the model. Qin & Eisner (2021) extends the idea of learning soft prompts and demonstrates that the implicit factual knowledge in language models was underestimated. Given that manually designing prompts can be cumbersome, automatically generating prompts gives a chance avoid intensive labor and enhance efficiency Schick et al. (2020); Schick & Schütze (2020). Gao et al. (2021) proposes to generate all prompt candidates and selectively incorporate them into each context using a refined strategy. Han et al. (2021) introduces a more efficient method to construct prompts with several sub-prompts that employs prompt tuning with rules without searching. Overall, prompt engineering is an efficient approach that helps bridge the gap between pre-training and fine-tuning.

### 2.5 CHAIN-OF-THOUGHT

Chain-of-Thought is a specialized tool designed for the task of multi-step reasoning and decision-making Wei et al. (2023). The traditional prompting method Brown et al. (2020a) performs poorly when it comes to tasks that require reasoning abilities. Inspired by the concept of using intermediate steps to solve reasoning problems Ling et al. (2017); Cobbe et al. (2021), the chain of thought method mimics a step-by-step thinking process and breaks down multi-step problems into intermediate steps, enabling the model to deduce more accurate results Narang et al. (2020). Additionally, Zhou et al. (2022b) address the challenge of dealing with tasks that are more complex than example prompts, and proposes the least-to-most prompting approach which breaks down complex problems into smaller and easier subproblems. Moreover, Wang et al. (2022) introduces self-consistency as a replacement for the greedy decoding algorithm, which samples and selects the most consistent reasoning paths to replace the greedy set.

## 3 METHOD

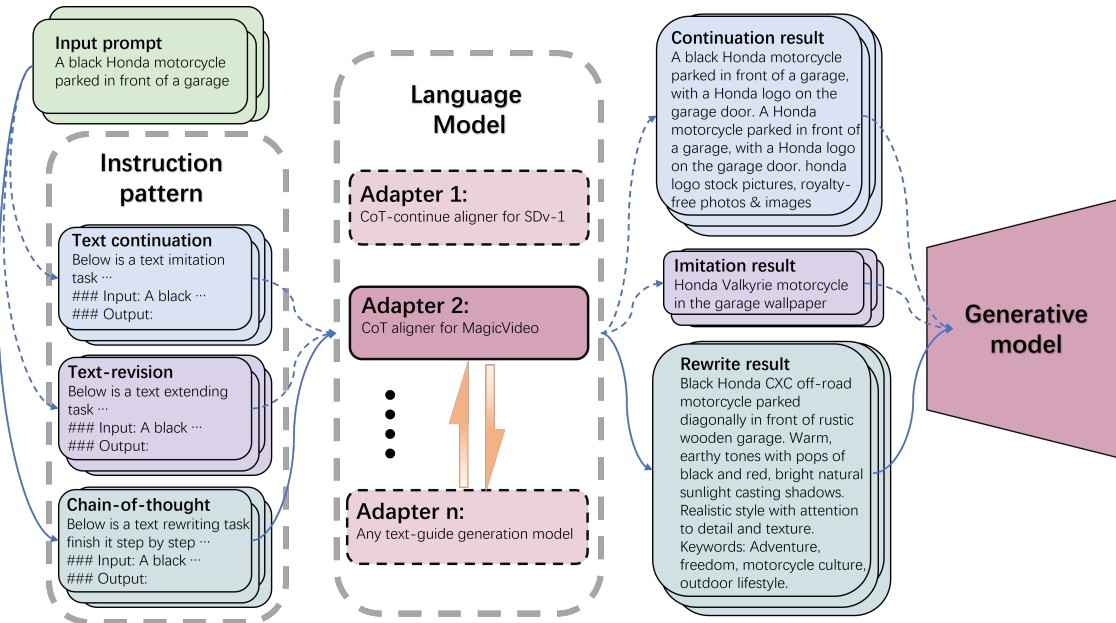

Figure 2: Pipeline of PromptCoT. (Left) We build three types of instruction patterns for training. (Middle) We utilize adapters for multi-task adaptation. (Right) Results of t-continue, t2t booster and PromptCoT.

### 3.1 OVERVIEW

Text-to-image diffusion models serve as an illustrative example for showcasing the functionality of PromptCoT. However, it is important to note that the same methodology can be extended and applied to other diffusion-based generative models, including text-to-video and various other domains. In the context of training text-to-image diffusion-based models, which involve image-text pairs and employ an iterative denoising process to reconstruct images based on corresponding prompts, our hypothesis posits that prompts aligned with high-quality images within the training set are more inclined to yield visually superior outputs. We randomly select 5 sets of 50 prompts corresponding to images with varying levels of quality from the Stable Diffusion training set, LAION Schuhmann et al. (2022), for image generation. The aesthetic score, an image quality metric introduced by Rombach et al. (2022), is used to represent the quality of individual images. As shown in Table 3.1, the generation performance is highly related to the prompts corresponding to the original image quality. For convenience, we refer to them as "**high-quality prompts**". In the following sections, we

Table 1: Comparison of Aesthetic Scores between Generated Images and Corresponding Training Images.

|                  | Aesthetic Score |     |     |     |
| ---------------- | --------------- | --- | --- | --- |
|                  | 4-5             | 5-6 | 6-7 | 7-8 |
| Training images  | 4-5             | 5-6 | 6-7 | 7-8 |
| Generated images | 5.2             | 5.5 | 6.1 | 6.3 |

explain the key components of PromptCoT, which is a prompt booster that can align input prompts with high-quality prompts in the training set, and in turn, improve generation performance.

### 3.2 ALIGNING PROMPT DISTRIBUTION WITH LLM

LLMs are extremely powerful tools that are capable of generating human-like language and completing tasks such as translation, summarization, question answering, etc. They are trained on massive amounts of text data and can learn from unstructured data to generalize to new tasks and domains. LLMs can also be fine-tuned on specific tasks with relatively small amounts of task-specific data, making them highly versatile. In this paper, we leverage this ability to align the distribution of high-quality prompts via fine-tuning a popular LLM LLaMA Touvron et al. (2023), on text continuation and revision tasks. To fine-tune LLaMA on text continuation, we use an instruction tuning template that includes incomplete text descriptions and a goal to provide a compelling continuation. The instruction tuning template is shown in Figure 3. We feed truncated text prompts placed in the *input* field to the LLM, supervised by the complete prompts. This enables the LLM to generate continuations containing more details.

Below is a text continuation task.
You will be given an incomplete text description.
Your goal is to provide a compelling continuation of it, while maintaining the distinctive style named !LAION-6plus-style!.
### Input:{input}
### Output:

Original Prompt=> A group of horses are grazing in the field.

Continuation=> A group of horses are grazing in the field. A lone tree stands in the center of the field. Storm clouds are entering from the left of the picture.

Figure 3: Template of text-continuation dataset (Up) and corresponding output (Bottom).

For text revision, we train the LLM to map human-like input texts to high-quality prompts. However, acquiring a large amount of human-written input text can be costly. Therefore, we leverage image captions from BLIP as a low-cost source of "human-like" input texts. The details of collecting and filtering data pairs are described in the later section. For training, we construct the instruction tuning template in Figure 4. The training pipeline is similar to continuation, but with the input being human-like prompts. As a result, we obtain a booster capable of performing revision tasks.

> Below is a text revision task.
> You will be given a text description.
> Try your best to rewrite it in a distinctive style named !LAION-6plus-style!.
> ### Input:{input}
> ### Output:

> Original Prompt=> People walking towards a parked bus on the street
>
> Revision=> In this picture taken on January 6, 2016 a man walks past a bus covered with snow at a bus stop in Kolomenskoye cemetery during heavy snowfall in Moscow. (Photo by Kirill Kudryavtsev/AFP Photo)

Figure 4: Template of text-revision dataset (Up) and corresponding output (Bottom).

### 3.3 ENHANCEMENT WITH CoT

Instruction tuning enables the LLM to add details and align text distribution, however, it tends to generate extraneous information that degrades performance. As such, we introduce the Chain-of-Thought (CoT) mechanism in the pipeline to address this issue. We set up five steps to make the LLM yield the expected production: (i) Extract key information from the original prompt, such as visual medium and main elements, (ii) Leverage the text-continuation model to append reasonable details, (iii) Extract additional concepts (for example, the color scheme) from the extended prompt and emphasize crucial concepts, (iv) With improved key information and crucial concepts, the LLM can generate a fluent prompt, remaining to be aligned, (v) Leverage the text-revision model to align prompts to the specific distribution. This mechanism extracts and amalgamates crucial information from the aligned continuation and revision, enabling reasonable inferences based on the contextual cues. As a result, a more comprehensive and nuanced final output is produced.

### 3.4 MULTI-TASK ADAPTATION

As the training set of different generative models can vary greatly, one approach to adapt to these new datasets is to fine-tune the entire LLM on the task-specific dataset. However, LLMs are typically models with billions of parameters, and allocating a dedicated LLM to each individual model proves impractical due to computational constraints. Moreover, there are plenty of text-to-image generative models trained on different datasets, and a single LLM cannot cover a diverse distribution of these datasets. As an alternative, we integrate adapters that facilitate dataset-specific adaptation, leveraging a shared pre-trained LLM as the foundation for this process. Adapters are lightweight modules that can be independently fine-tuned and subsequently added to the base model. Keeping adapters instead of the whole model significantly reduces memory usage, while enabling the adaptation of the LLM to different datasets.

### 3.5 DATASET PREPARATION

We build three types of datasets: text-continuation, text-revision, and text-CoT.

**Text-continuation dataset.** To create this dataset, we filter high-quality prompts from the training data of existing generative models, using criteria such as high CLIP similarity and proper length. In the case of the LAION dataset, we also consider aesthetic scores to ensure a higher quality of prompts. Once high-quality prompts are identified, we truncate a portion of the text, with the remaining front part assigned as input data. The LLM is then trained to generate the missing information and complete the text. This process enables the LLM to learn how to effectively continue text prompts in a manner that is consistent with the style and context of the original text.

**Text-revision dataset.** The dataset consists of human-like texts and corresponding high-quality prompts which are described in the text-continuation dataset. To acquire human-like prompts, we leverage BLIP and CLIP-interrogator for image captioning. Furthermore, we calculate the text distance with the text encoder of CLIP, ensuring a score greater than 0.4 to guarantee semantic relevance between the two prompts.

**Text-CoT dataset.** We use GPT-3.5-Turbo to build a task-specific dataset. Initially, we design a step-by-step interaction with GPT-3.5-Turbo to extract and guide the prompt booster to finish the

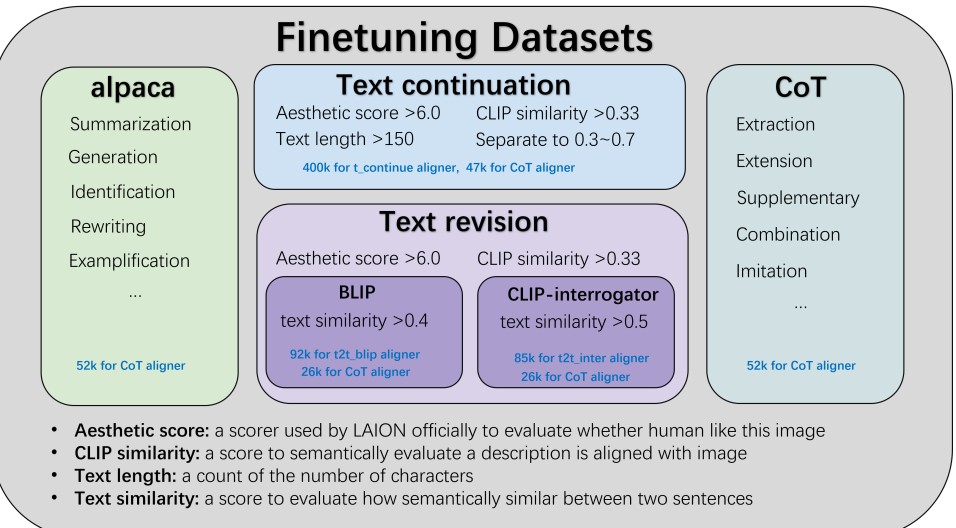

Figure 5: Composition of fine-tuning tasks including text-continuation, text-revision, text-CoT, and self-instruction of Alpaca.

alignment task, due to the fact that CoT is still difficult for alpaca with a simple finetuning on datasets above. Following the alpaca's thought, 52k pairs are all generated from gpt-3.5-turbo.

# 4 EXPERIMENTAL RESULTS

In this section, we first introduce the details on the datasets, pre-trained models, and the training hyperparameters used for all our experiments in Section 4.1. Then we demonstrate the results of applying PromptCoT to text-to-image and text-to-video pre-trained generative models in Section 4.2 and Section 4.3 respectively.

Figure 6: Generated images from prompts refined by different aligners. (a) and (h) show the images generated with the original text prompts. (b-g) and (i-n) denote the images generated with text prompts refine by 't-continue', 't2t-blip','t2t-inter','davinci','CoT_d', and 'CoT' respectively.

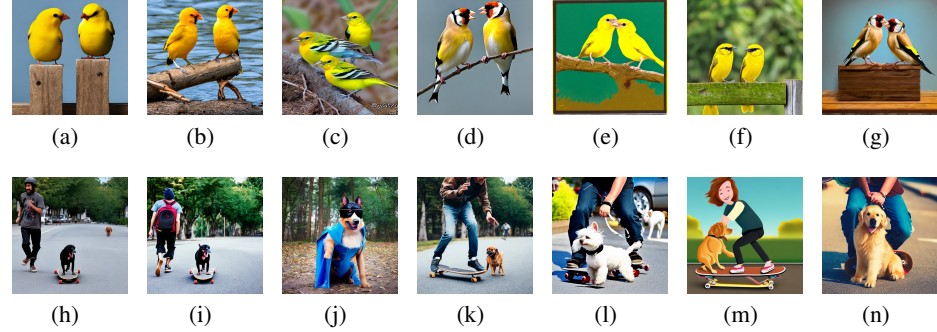

## 4.1 SETUP

Dataset. For training, we build Text-revision and Text-continuation dataset from LAION-aes6plus Schuhmann et al. (2022), and Text-CoT dataset with the help of GPT-3.5-turbo. LAION-aes6plus is the subset of LAION, containing 12M image-text pairs with predicted aesthetics scores of 6 or higher. As a supplement, we also train with Text-revision, Text-continuation, and Text-CoT datasets from the WebVid-10M dataset Bain et al. (2022) for video generation. For evaluation, we conduct experiments on COCO Lin et al. (2014) validation set and MSR-VTT Xu et al. (2016) for FID, FVD, aesthetic score, CLIP score, and PickScore.

Models. The pre-trained LLaMA-7B is used as the base model and we employ the adapter design outlined in Zhang et al. (2023) to facilitate multi-task adaptation. Two versions of Stable Diffusion Rombach et al. (2022), v1.4 and v2.1, are used for image generation. MagicVideo Zhou et al. (2022a) is used for video generation.

Implementation Details. We finetune the LLaMA following alpaca's Taori et al. (2023) strategy and instruction pattern, which has been verified powerful for text generation tasks. We validate the viability of our two initial ideas by finetuning three task-specific LLaMA for prompt refining works shown in experiments 4.1. One is trained on the self-constructed text-continuation dataset while the other two are trained on two types of text-revision dataset. While combining such basic methods by CoT, we include a dataset from alpaca, a subset of the text-continuation dataset, and the text-revision dataset with higher text similarity and the CoT dataset as a whole. We evaluate our alignment work on three diffusion models and on different parameters. Furthermore, we evaluate the portability of promptCoT through an adapter by comparing its performance with the fully-finetuned model.

Table 2: **Text-to-image generation performance.** We evaluate the generation performance on Stable Diffusion v1.4 and v2.1 on key metrics including aesthetic score, FID, IS, CLIP score and PickScore.

| Generation Model | Booster | Aesthetic Score | FID | IS | CLIP Score | PickScore (avg/recall) |
|---|---|---|---|---|---|---|
| SD v1.4 ddim step=50 scale=7.0 | baseline | 5.40 | 59.15 | 39.13 ± 0.84 | 0.268 | 27.3%/35.7% |
| | t-continue | 5.54 | 44.66 | 35.81 ± 0.96 | 0.290 | 39.5%/61.5% |
| | t2t-blip | 5.62 | 40.77 | 38.56 ± 0.77 | 0.293 | 51.4%/77.5% |
| | t2t-inter | 5.44 | 55.76 | 41.00 ± 1.17 | 0.271 | 34.3%/49.0% |
| | cot_d | 5.64 | 49.58 | 37.43 ± 0.94 | 0.289 | 40.6%/62.2% |
| SD v2.1 ddim step=50 scale=7.0 | baseline | 5.60 | 58.02 | 37.51 ± 1.00 | 0.266 | 29.4%/41.7% |
| | t-continue | 5.70 | 45.62 | 34.44 ± 0.71 | 0.287 | 44.3%/69.9% |
| | t2t-blip | 5.79 | 40.59 | 37.38 ± 1.08 | 0.292 | 56.3%/82.5% |
| | t2t-inter | 5.64 | 54.93 | 38.60 ± 0.85 | 0.269 | 37.1%/55.6% |
| | cot_d | 5.78 | 50.41 | 34.88 ± 0.95 | 0.290 | 42.9%/66.2% |
| SD v2.1 ddim step=250 scale=12.0 | baseline | 5.60 | 58.17 | 36.37 ± 0.81 | 0.267 | - |
| | t-continue | 5.64 | 46.59 | 33.29 ± 0.68 | 0.287 | - |
| | t2t-blip | 5.76 | 40.89 | 36.16 ± 0.84 | 0.292 | - |
| | t2t-inter | 5.64 | 55.37 | 38.10 ± 1.16 | 0.269 | - |
| | cot_d | 5.75 | 50.41 | 34.88 ± 0.94 | 0.290 | - |

Table 3: **Text-to-image generation performance with adapters.** We fine-tune adapters by 5 epochs and compare them with fully fine-tuned Alpaca. Model with adapters achieves comparable results.

| Generation Model | Booster | Aesthetic Score | FID | IS | CLIP Score |
|---|---|---|---|---|---|
| Adapter epochs = 5 | t-continue | 5.69 | 48.00 | 35.8 ± 0.57 | 0.283 |
| | t2t-blip | 5.70 | 46.86 | 38.0 ± 0.66 | 0.289 |
| | t2t-inter | 5.64 | 56.28 | 39.0 ± 0.64 | 0.269 |
| | cot_d | 5.85 | 51.06 | 31.8 ± 0.65 | 0.251 |

## 4.2 TEXT-TO-IMAGE EVALUATION

The COCO Lin et al. (2014) validation set is the standard benchmark for evaluating text-to-image models. The key automated performance metrics used are FID to measure image fidelity, CLIP score, PickScore to measure image-text alignment, aesthetic score Murray et al. (2012) to predict the aesthetic quality, and Inception Score (IS) to evaluate the diversity. We utilize two versions of Stable Diffusion for image generation with prompts from COCO and our PromptCoT. Table 4.1 presents the evaluation results for each metric with different single-function boosters including t-continue, t2t-blip, and t2t-inter, as well as a baseline. The results show that incorporating the alignment method proposed in our paper consistently improved the generated image quality across all metrics compared to the baseline. Among the single-function boosters, the t2t-blip booster demonstrates the best performance, as it is able to achieve alignment to a greater extent. For example, it transfers "Boxes

Table 4: **Text-to-image generation performance.** We compare finetuned CoT aligner and davinci-003 model from OpenAI. All metrics are evaluated on a subset of the COCO validation dataset which contains 1k images.

| Booster | Aesthetic Score | CLIP Score | PickScore |
|---|---|---|---|
| baseline | 5.62 | 0.231 | 16.8%/26.1% |
| davinci | 5.69 | 0.277 | 26.0%/47.5% |
| tcontinue | 5.72 | 0.285 | 37.8%/66.2% |
| t2t_blip | 5.80 | 0.293 | 50.6%/81.5% |
| t2t_inter | 5.66 | 0.269 | 30.7%/52.5% |
| cot_d | 5.79 | 0.291 | 34.9%/59.5% |
| **cot** | **5.80** | **0.293** | 36.4%/59.0% |

of fruit displayed at an open-air market" to "A view of stalls selling fruit at the Harare International Market in Harare, Zimbabwe" by rephrasing the expression and adding reasonable details. In contrast, the t2t-inter booster, which has a similar function to t2t-blip, shows inferior performance, although it still outperforms the baseline. This could be due to the CLIP-interrogator used to create the text-revision dataset introducing irrelevant entities. Furthermore, we test with different factors of classifier-free guidance to prove the generality of our PromptCoT. Varying the scale of classifier-free guidance results in consistent performance.

## 4.3 TEXT-TO-VIDEO EVALUATION

In addition, we experiment with the text-to-video task to demonstrate the effectiveness of our approach. We employ PromptCoT on the WebVid-10M dataset Bain et al. (2022). Then, we finetune the LLaMA model following alpaca's Taori et al. (2023) strategy and refine prompts from MSR-VTT with the fine-tuned model. We use MagicVideo Zhou et al. (2022a) as the base model to test the effectiveness of our prompts. The results are shown in Table 5. The results indicate that the PromptCoT are effective in enhancing the quality of the generated videos compared to the baseline. Among the boosters, the PromptCoT better aligns the prompts and achieves the best performance overall. For cot_d, we generate 21k data with the help of GPT-3.5-turbo. Similar to text, we utilize a chain of five questions to generate the expected production, but with subtle differences to encourage GPT-3.5-turbo to generate more video-related features, e.g., movement. Similar to text generation, we adopt a chain of five questions to generate the expected production for video prompts. However, there are subtle differences in the question prompts to encourage GPT-3.5-turbo to incorporate more video-related features, such as movement, into its generated content. For example, "a large passenger jet flying in the sky at sunset" can be refined to "Boeing 747 flying across a vibrant sunset backdrop in a captivating, cinematic 4K video. Slowly gaining altitude with wings tilting slightly, this footage captures the plane's majesty". The scores of cot_d will be included in the supplementary material.

Table 5: **Text-to-video generation performance.** We evaluate the generation performance on MagicVideo on key metrics including FID, FVD, and CLIP score.

| Model | Dataset | Booster | FID | FVD | CLIP Score |
|---|---|---|---|---|---|
| MagicVideo | MSR-VTT | baseline | 36.5 | 998 | 0.284 |
| | | t-continue | 33.2 | 951 | 0.296 |

## 5 CONCLUSION

In this paper, we present PromptCoT, an innovative system designed to autonomously enhance the quality of prompts used in diffusion-based generative models, which are critical for high-fidelity visual content generation. PromptCoT leverages pre-trained Large Language Models (LLMs) and a unique Chain-of-Thought (CoT) mechanism to refine prompts, thereby improving the alignment between the original and refined prompts. To balance computational efficiency, we employ adapters to allow for efficient adaptation to new datasets or models. Our evaluations demonstrate that PromptCoT can achieve superior performance compared to the baselines.

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
