# PromptCoT: Align Prompt Distribution via Adapted Chain of Thought (Supplementary material)

## A   Overview

This document serves as supplementary material to the main paper. We present additional implementation details in Section B, including the construction of datasets, fine-tuning settings, and an introduction to evaluation metrics. Section C contains additional experimental results, while Section D discusses the ablation study on CoT dataset and adapters. Furthermore, we include extra visualization examples in Section E. We also address the limitations and societal impact of our work in Section F.

## B   Additional Implementation Details

**Data collection.**   We first explore how the length of the text descriptions impacts the generation performance of the model. Figure 1 displays the distribution of text length in the LAION dataset Schuhmann et al. (2022), revealing that the majority of text descriptions fall within the range of 10 to 150 characters. To facilitate distinct analysis, the dataset is divided into three separate groups, each consisting of 20,000 data samples. The first group, named *short-cap*, encompasses captions with a length of less than 40 characters. The second group, referred to as *mid-cap*, comprises captions exceeding 90 characters but falling short of 110 characters. Finally, the third group, denoted as *long-cap*, includes captions surpassing 150 characters. The intentional avoidance of consecutive length ranges ensures clear differentiation between the groups, allowing for ease of distinction. Utilizing a pre-trained latent diffusion model, three sets of images are generated based on the text descriptions from the respective groups. The calculated mean aesthetic scores Rombach et al. (2022) for each group are as follows: 6.01 for *short-cap*, 6.03 for *mid-cap*, and 5.99 for *long-cap*. Furthermore, the Fréchet Inception Distance (FID) Heusel et al. (2017) is computed, resulting in values of 13.1 for *short-cap*, 9.4 for *mid-cap*, and 10.8 for *long-cap*. Notably, no significant impact of text length on the quality of the generated images is observed. Consequently, a uniform sampling strategy is employed for all sub-datasets utilized throughout the paper.

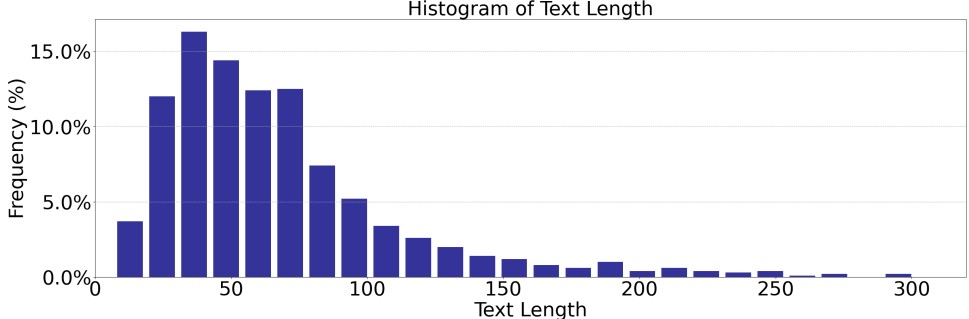

Figure 1: The distribution of text lengths in the LAION dataset.

**Training settings.**   All experiments are based on pre-trained LLaMA-7B Touvron et al. (2023), an open-sourced Large Language Model with seven billion parameters. The fine-tuning process of each

aligner follows Taori et al. (2023); Wang et al. (2022a) using $8\times$A100-80GB GPUs, which takes three hours until converge. More specifically, we set 2e-5 for the learning rate, 0.0 for weight_decay, 0.03 for warmup_ratio, and cosine decay for the learning rate schedule. For all one-step aligners, including text continuation, text imitation, and direct aligner with training dataset from CoT, the max sequence length is set to 512 while the batch size is 2 and gradient accumulation steps are 8. For CoT aligners, the max sequence length is set to 1500 while the batch size is 1 and the gradient accumulation steps are 2.

**Adapter setting.** In PromptCoT, we add adapter layers following Gao et al. (2023). For all aligners, we set the number of adapter layers to 30 with each length of 10, initial learning rate to 9e-3, weight_decay to 0.02 and 5 epochs within 2 warming up epochs. For all one-step aligners, including text continuation, text imitation, and direct aligner with the training dataset from CoT, the max sequence length is set to 512 while batch size is 8. For PromptCoT aligners, the max sequence length is set to 1500 while batch size is 1. The use of adapter significantly reduces memory cost since it takes $n \times 26GB$ for $n$ finetuned aligners but only $26GB + n \times 4.8MB$ for $n$ aligners with adapters.

**Evaluation Metrics.** We evaluate the generation performance with Fréchet Inception Distance (FID) Heusel et al. (2017), Inception Score (IS) Salimans et al. (2016), CLIP score Radford et al. (2021), Aesthetic Score Rombach et al. (2022) and PickScore Kirstain et al. (2023). The definitions of FID, IS, and CLIP score are strictly following previous worksHeusel et al. (2017); Salimans et al. (2016); Radford et al. (2021); Rombach et al. (2022); Kirstain et al. (2023). We here give more detailed explanations of Aesthetic Score and PickScore in this paragraph.

*Aesthetic Score* is calculated with a pre-trained aesthetics predictor provided by LAION Schuhmann et al. (2022). It also has been used for data filtering of recent popular latent diffusion models Rombach et al. (2022). It is designed based on CLIP ViT/14 with an extra linear layer at the top of the model. The model is optimized to predict the ratings collected from people's answers to questions such as "How much do you like this image on a scale from 1 to 10?". In this paper, we use the aesthetic score to show that after being refined by our prompt aligner, generative models can create images that human regards as amusing.

*PickScore* Kirstain et al. (2023) is a scoring function trained over Pick-a-Pic by combining a CLIP-style model with a variant of InstructGPT's Ouyang et al. (2022) reward model objective whose goal is to predict human preferences. We use PickScore to construct two kinds of evaluation metrics to represent how humans like the generated image. Each time we input a group of generated images led by prompts refined from our different aligners and the prompt refined from the aligner being evaluated. The average PickScore is the probability that a human is predicted to prefer the image generated by the input prompt among this group of images, while the recall PickScore is the rate that predicted human reaction is preferring the corresponding image.

## C Additional Experiments

### C.1 PickScore for Adapter

We provide additional PickScore results of aligners with adaptation in Table C. Experiments indicate that all aligners consistently improve this metric compared to the baseline.

Table 1: **Text-to-image generation performance of aligners with adaptation.**

| Base Model | Aligner | PickScore(%) (Average/Recall) |
|---|---|---|
| Adapter | baseline | 27.3/37.3 |
| | t-continue | 41.1/67.9 |
| | t2t-blip | 42.5/66.7 |
| | t2t-inter | 33.8/48.7 |
| | cot_d | 41.9/66.2 |

Table 2: **Text-to-image generation performance of fully fine-tuned aligners.**

| Generator | Aligner | PickScore(%) (Average/Recall) |
|---|---|---|
| SD v2.1 ddim step=250 scale=12.0 | baseline | 29.3/41.9 |
| | t-continue | 44.7/70.7 |
| | t2t-blip | 56.4/83.2 |
| | t2t-inter | 37.3/56.4 |
| | cot_d | 41.0/64.4 |

## C.2 PICKSCORE FOR STABLE DIFFUSION V2

Table 2 presents additional PickScore Kirstain et al. (2023) results for the generation performance of various aligners on the COCO Lin et al. (2014) validation set. The experiments are conducted using Stable Diffusion v2.1. Our results show that all aligners significantly outperform the baseline on this metric.

## D ABLATION STUDY

### D.1 TRAINING PROMPTCOT EXCLUSIVELY WITH COT DATASET

We conducted the ablation study to compare the performance of the full-pipeline PromptCoT aligner, *cot*, with several variants on a subset of the COCO Lin et al. (2014) validation dataset consisting of 1,000 images. The variants included *cot_d*, which is an aligner trained exclusively on the results of the final step (step 5) to accelerate inference. The variants also include *cot_only*, which is trained without datasets of Alpaca Taori et al. (2023), text continuation, and text imitation, solely on the CoT dataset to accelerate training. Our experiments (Table 3) indicate that although these more efficient variants have a subtle impact on marginal aspects, they still deliver impressive final performance.

Table 3: **Text-to-image generation performance on different CoT aligners.** All metrics are evaluated on a subset of the COCO Lin et al. (2014) validation dataset consisting of 1,000 images. Images are generated by Stable Diffusion with corresponding prompts under the same conditions.

| Aligner | Aesthetic Score | CLIP Score | PickScore (%) (Average/Recall) |
|---|---|---|---|
| baseline | 5.62 | 0.231 | 28.4/40.7 |
| cot_d | 5.79 | 0.291 | 47.0/**65.1** |
| cot_only | **5.80** | **0.293** | 43.2/59.5 |
| cot | **5.80** | **0.293** | **47.2**/64.4 |

### D.2 PROMPTCOT WITH ADAPTER

Table 4: **Text-to-image generation performance with adaptation.** PromptCoT with adaptation achieves comparable results compared to the fully fine-tuned counterpart.

| Base Model | Aligner | Aesthetic Score | FID | CLIP Score |
|---|---|---|---|---|
| Adapter | baseline | 5.60 | 58.02 | 0.266 |
| | cot_d | **5.85** | 51.06 | 0.251 |
| | PromptCoT | 5.80 | **46.54** | **0.291** |

We further conduct a complementary evaluation of full-pipeline PromptCoT with the adaption approach on COCO validation dataset with 25,000 images in Table 4. Experiments indicate that

adaptation achieves comparable performance on Aesthetic Score and improvement on FID and CLIP Score, compared to the fully fine-tuned counterpart.

### D.3 COMPARISON BETWEEN PROMPTCOT AND HUMAN-REFINED PROMPTS

To compare the capability of refining prompts between PromptCoT and human beings, we first collect a set of text prompts from the captions of COCO dataset. We then invited a group of 30 research volunteers to refine the collected prompts to improve the image generation quality. The volunteers are all specialized in deep learning algorithms and are thus expected to perform well on this task. The findings are succinctly presented in Table 5. Upon careful examination, it is evident that humans possess the ability to modify prompts to achieve better content alignment between the text descriptions and the generated images, resulting in an improved CLIP score. However, it should be noted that there is a slight decrease in aesthetic scores when employing this approach. Conversely, PromptCoT demonstrates its capability to generate prompts that enhance not only the aesthetic score but also the CLIP score and PickScore, surpassing human performance by a significantly larger margin.

Table 5: **Comparison to human-refined promtps. We evaluate the generation quality on Aesthetic Score Rombach et al. (2022), CLIP Score Radford et al. (2021) and PickScore Kirstain et al. (2023).**

| Aligner | Aesthetic Score | CLIP Score | PickScore(%) (Average/Recall) |
|---|---|---|---|
| Baseline | 5.68 | 0.23 | 33.2/39.1 |
| Human | 5.62 | 0.27 | 48.1/58.2 |
| PromptCoT | **5.77** | **0.30** | **57.5/73.6** |

## E ADDITIONAL VISUALIZATION

### E.1 IMPACTS OF PROMPTS IN TRAINING DATA ON GENERATION PERFORMANCE

Our empirical findings indicate a positive correlation between the quality of prompts associated with high-quality images in the training dataset and the generation of superior images when applied to pre-trained latent diffusion models. This relationship is visually represented in Figure 2. Figure 2 portrays an instance of a text-image pair characterized by low visual quality, prominently displayed in the top-left corner and highlighted in orange. Consequently, the resulting generated images derived from such prompts exhibit a corresponding decline in visual quality. Conversely, the last two rows of Figure 2 present a contrasting scenario where text prompts sourced from high-visual-quality training samples yield images of commendable visual quality.

### E.2 IMPACTS OF PROMPTCOT COMPARED TO ONLINE USERS

In this section, we utilize prompts collected from an online database Wang et al. (2022b), where users share their self-generated prompt-image pairs. We also verify the effectiveness of PromptCoT on those real-world prompts. The results are shown in Figure 4. The left column shows the images generated with the original prompt used by the public and the right column shows the images generated with the refined prompt by PromptCoT. The original prompt and the refined prompt are also listed under the corresponding image pairs. It is essential to highlight that the quality of the generated images cannot be attributed solely to the prompt's length. Even when users provide detailed descriptions, the generated images may still fall short of expectations. For example, in the first row in Figure 4, the online user attempts to depict a construction worker in a construction field by providing unorganized key concepts. However, the resulting generation exhibits flaws in the worker's clothing, eyes, and background, indicating a lack of coherence and quality. In the second-row pairs, the user-generated image lacks the "full body" concept, leading to a partial representation of the prompt. In the bottom-row pairs, the user's prompt for generating the well-known character "Rocket Raccoon" exhibits unrealistic body proportions. In each of these instances, the utilization of PromptCoT yields

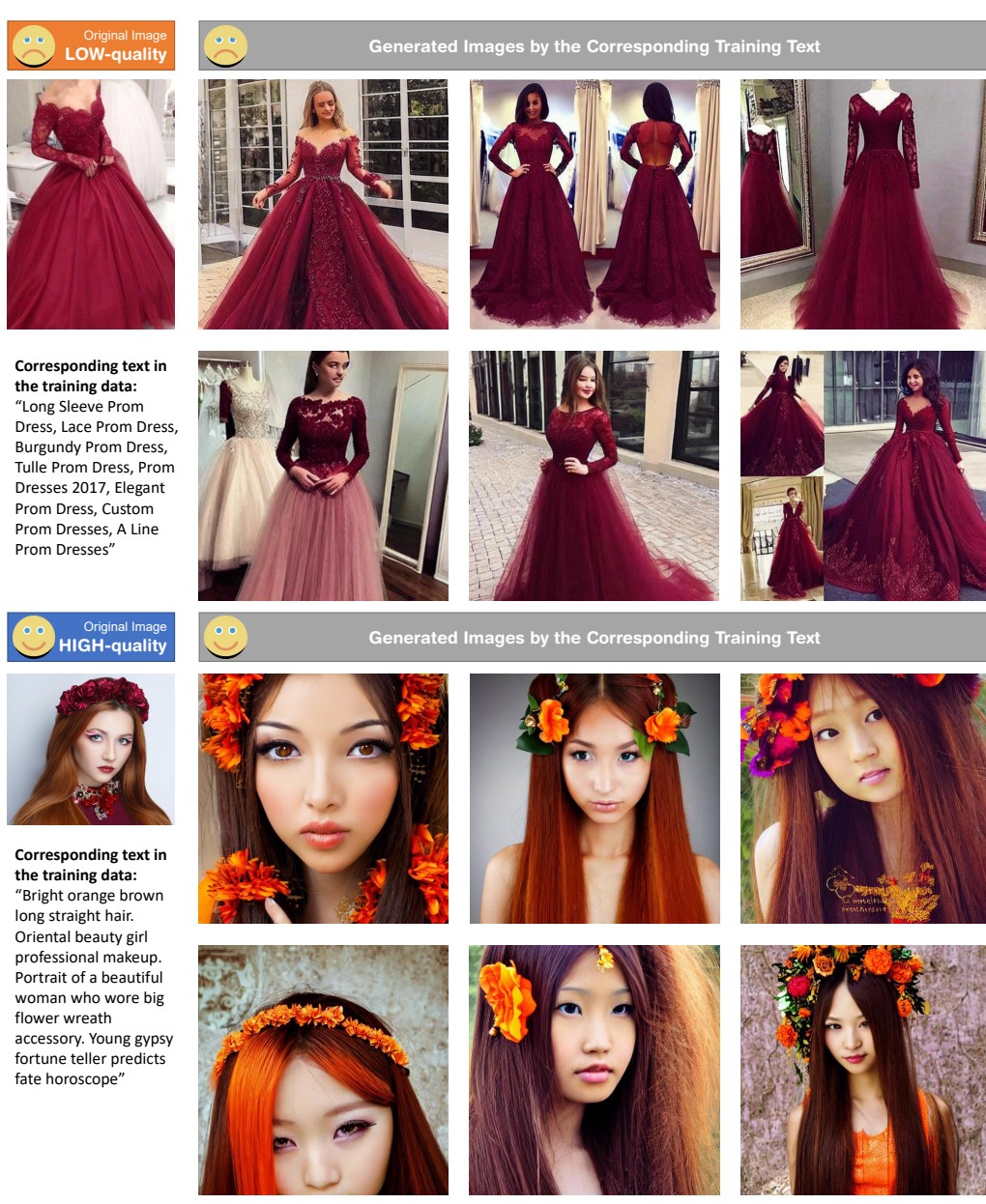

Figure 2: "Low-quality prompt" refers to the text in the training set whose corresponding image **(left)** has low quality. **(Up)** Images generated by a low-quality prompt. "High-quality prompt" refers to the text in the training set, and whose corresponding image has high quality. **(Bottom)** Images generated by a high-quality prompt.

a noteworthy enhancement in the quality of generated outputs. This improvement is achieved through the process of prompt re-writing, which ensures a more effective alignment with the training text data. As a result, the generated images exhibit a heightened level of fidelity and aesthetics, thereby attaining a closer resemblance to the intended expectations.

## E.3    VISUALIZATION OF DIFFERENT ALIGNERS

In Figure 5, we provide a detailed visual comparison of images generated using the original prompt and those refined with different aligners (tcontinue, t2t_blip, t2t_inter, cot_davinci, cot_d, and

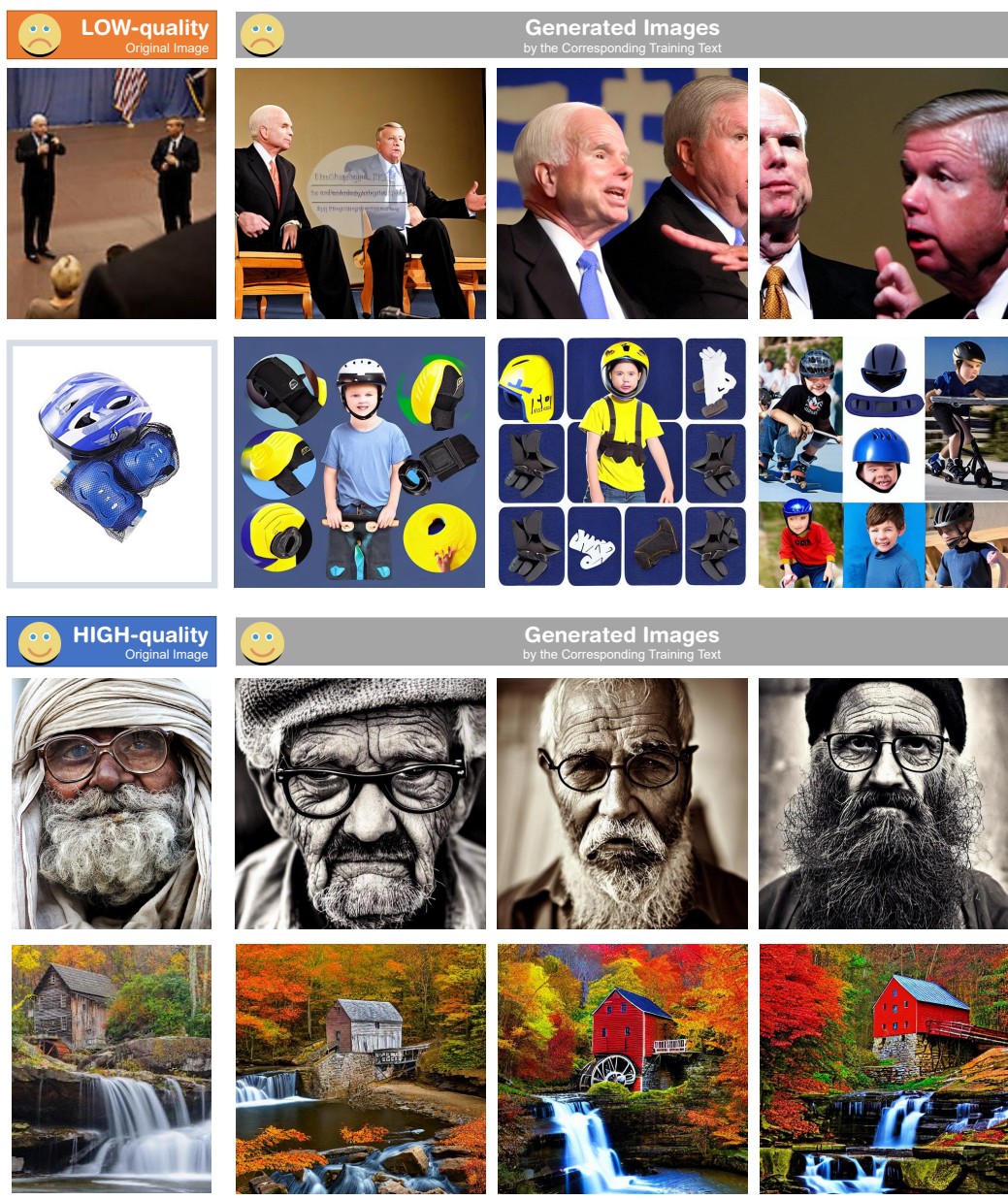

Figure 3: More examples of images generated by low/high-quality prompts.

PromptCoT). We have highlighted inconsistencies between the prompt and the images within the figures, accompanied by annotations below each image. It is noteworthy that not only do the images generated using PromptCoT exhibit superior quality, but they also display a better alignment with the textual contents. For instance, in the top-row images generated from the prompt "A surfer on a whiteboard riding a small wave," PromptCoT stands out by effectively capturing all the desired elements, while others may struggle to interpret the prompt accurately with all key concepts.

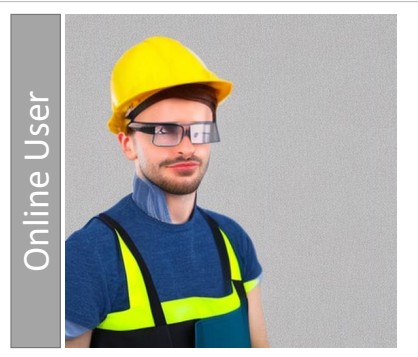
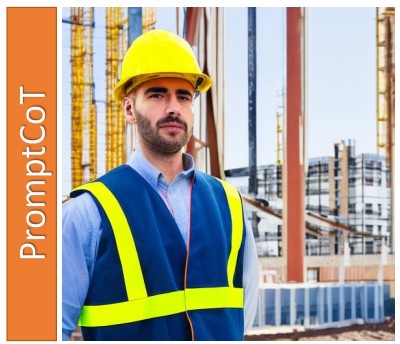

**Left by the online user:** "construction worker. portrait. bauhaus, angular, geometric, symmetrical. yellow color theme. construction background."
**Right by PromptCoT:** "Professional Construction Worker in Yellow Hard Hat and Reflective Vest Standing in Front of Bauhaus-Inspired Construction Site"

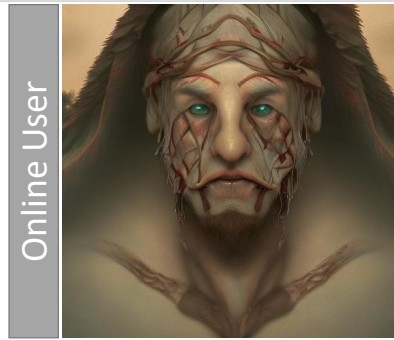
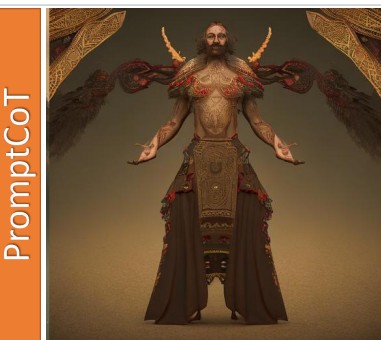

**Left by the online user:** "a fantasy character Drekavak at Proto-Slavic mythology. The soul of a dead unbaptized infant, has the ability to scream eerily.. Full body, detailed and realistic,4k, top-artstation, inspired blizzard games, octane render"
**Right by PromptCoT:** "Fantasy character Drekavak, inspired by Proto-Slavic mythology, depicted in a dramatic pose with outstretched arms, wearing intricate clothing adorned with Proto-Slavic designs. Rendered in 4k resolution using Octane render, with a dark and foreboding background to create an eerie atmosphere."

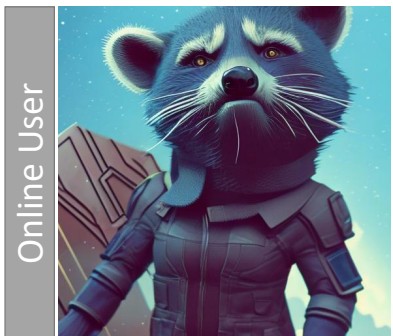
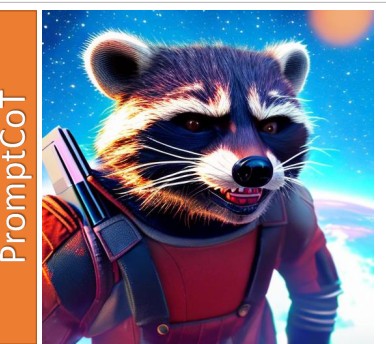

**Left by the online user:** "rocket raccoon, space background, close up, quint buchholz, wlop, dan mumford, atgerm, liam brazier, peter mohrbacher, raw, featured on artstation, octane render, cinematic, rugged, intricate, 8 k"
**Right by PromptCoT:** "Rocket Raccoon in Space" - An AI-generated digital painting featuring Rocket Raccoon in a close-up shot against a rugged and intricate space background with stars and planets. Rendered in Octane Render with 8K resolution, the image boastsa vibrant color scheme. dramatic lighting, and a realistic style. Inspired by arists such as Quint Buchholz, WLOP, Dan Mumford, Artgerm, Liam Brazier, and PeteMohrbacher, this cinematic image is sure to be a standout on ArtStation."

Figure 4: Comparison between the online users and PromptCoT. Images are placed in pairs of (left) the online user and (right) PromptCoT.

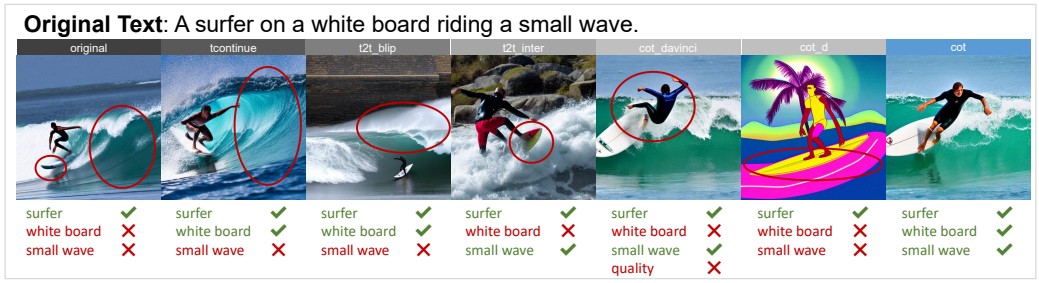

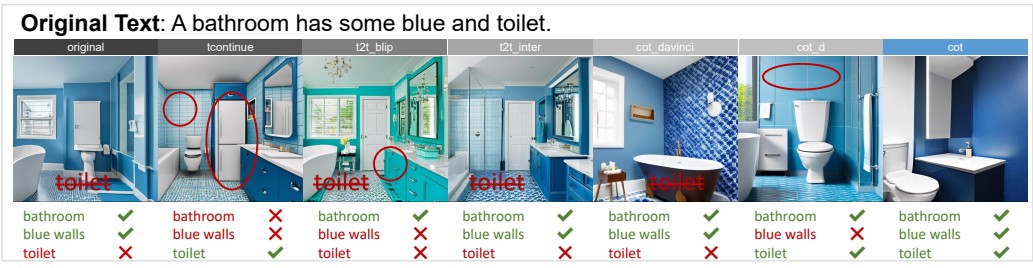

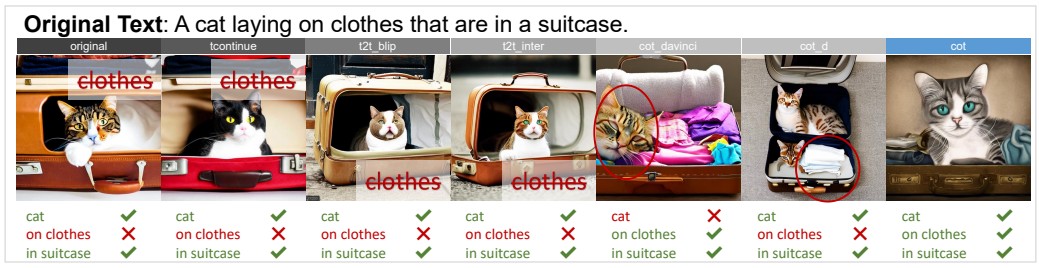

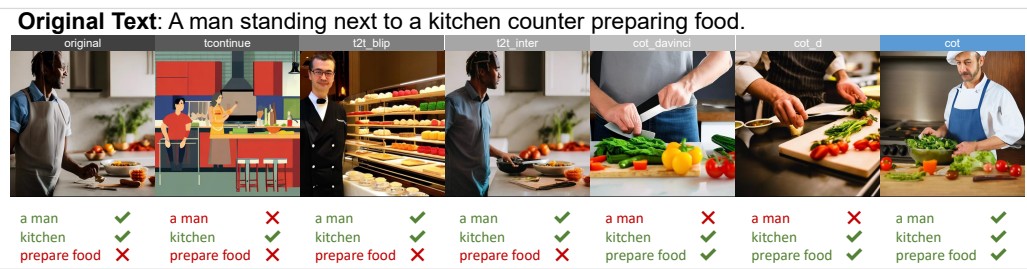

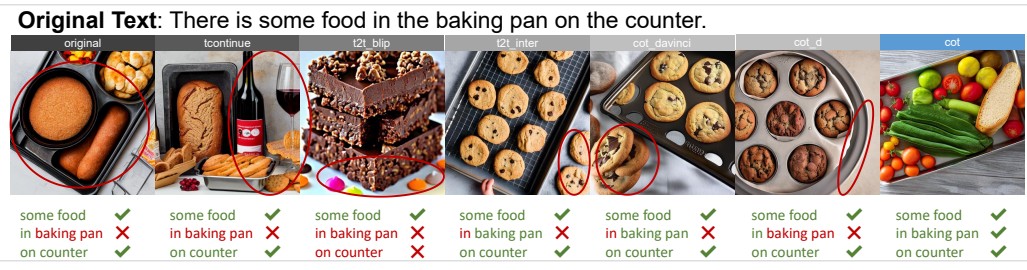

Figure 5: From left to right, images are generated via original prompts and prompts refined by tcontinue, t2t_blip, t2t_inter, cot_davinci, cot_d, and PromptCoT, respectively.

## F  LIMITATIONS AND SOCIETAL IMPACT

**Limitations**   While PromptCoT is able to enhance the generation performance of generative models by a significantly larger margin, the extent of this enhancement is reliant on the underlying capabilities of the pre-trained generative models. Additionally, if the prompts provided to the generative models are already of high quality, the further improvements brought by PromptCoT would also be limited.

**Societal Impact**   We believe that PromptCoT is a versatile approach that can help users to improve the quality of the generation performance by a large margin on various generative applications, reducing the re-generation process and thus reducing the emission of greenhouse gases. Moreover, with lightweight adaptation, PromptCoT can be applied to multiple tasks within negligible memory overhead, providing a highly efficient once-for-all approach for industrial deployment. However, in this study, we only evaluated the effectiveness of PromptCoT in enhancing visual quality-related performance and did not address longstanding concerns related to privacy, security, and copyright issues in the field. In future research, we will explore the effectiveness of PromptCoT in addressing these concerns and ensuring the safety of generated content, while maintaining high-quality generation.