# OpenReview forum: "PromptCoT: Align Prompt Distribution via Adapted Chain-of-Thought"
_ICLR.cc/2024/Conference — ICLR 2024 Conference Withdrawn Submission_

### Official Review · Reviewer_YkZW · 2023-10-15

**Soundness:** 3 good
**Presentation:** 3 good
**Contribution:** 3 good
**Rating:** 6
**Confidence:** 2

**Summary:**

The paper introduces "PromptCoT," a novel approach to enhance the quality of prompts used in diffusion-based generative models for text-to-image generation. The authors address the challenge of prompt engineering, which typically requires expertise and can be challenging for inexperienced users. PromptCoT leverages a pre-trained Large Language Model (LLM) to autonomously refine prompts for users, improving the performance of generative models. The refinement process involves fine-tuning the LLM on a curated dataset of high-quality visual content descriptions to better capture the distribution of training texts. The paper also introduces the Chain-of-Thought (CoT) mechanism to align original text prompts with refined versions. Additionally, the authors incorporate adapters for dataset-specific adaptation, promoting computational efficiency.

**Strengths:**

- This paper is written in a clear and easily comprehensible manner, making it easy for readers to follow.
- The paper is well-written, with no apparent typos or grammatical errors. However, thorough proofreading is always recommended to ensure clarity and professionalism in scientific writing.
- PromptCoT introduces an innovative approach to address the critical issue of prompt quality in generative models. The use of fine-tuning and the CoT mechanism provides a unique solution to enhance the performance of diffusion-based models.

**Weaknesses:**

see questions.

**Questions:**

- Some prior works, such as SUR-adapter [1] and LLM-grounded Diffusion [2], have also explored the alignment between large language models and input prompts with the text distribution. It would be valuable for the authors to discuss the similarities and differences between their proposed method and these existing works. This discussion could be incorporated into the introduction or related work section.

- What does the magnitude of the Aesthetic Score differences signify? For example, in Table 2, is there a substantial distinction between Aesthetic Scores of 5.64 and 5.78? What range of values is considered significant? Additionally, could visualizations be provided to help interpret the differences in Aesthetic Scores?

- It's noted that CLIP scores are used in this paper. Given that the diffusion model used in this paper already incorporates CLIP's text encoder, is it still meaningful to use CLIP scores for evaluation? Since the generated images are inherently based on CLIP's information, it's possible that their CLIP scores may not accurately reflect the level of alignment between text and images.

- This paper includes various hyperparameters such as "Aesthetic score > 6.0," "CLIP similarity > 0.33," and "text similarity > 0.4," which require thorough discussion or at the very least, appropriate references.

- The author needs to clarify the above questions. If these issues are addressed, I will consider these clarifications along with feedback from other reviewers in deciding whether to raise my score.

[1] SUR-adapter: Enhancing text-to-image pre-trained diffusion models with large language models, ACM MM

[2] LLM-grounded Diffusion: Enhancing Prompt Understanding of Text-to-Image Diffusion Models with Large Language Models, arxiv

---

### Official Review · Reviewer_BjAS · 2023-10-28

**Soundness:** 2 fair
**Presentation:** 2 fair
**Contribution:** 1 poor
**Rating:** 3
**Confidence:** 3

**Summary:**

The paper introduces an automatic prompt engineering method based on Chain of Thoughts (CoT) that tries to use text resembling the high quality images in the training set to compose prompts that are more likely to generate high quality images for diffusion-based generative models. The method works by providing a truncated prompt and asking LLM to complete it with more details. To avoid generating extraneous information, authors utilize CoT to use text-continuation model to only add reasonable additional information based on the extracted critical visual components from the original prompt. Authors report experimental results of evaluating their method vs. a baseline and a few other alternatives for text to image and text to video generation.

**Strengths:**

- The core idea of using CoT and aligning prompt with text associated with high quality training instances make sense.
- CoT formulation consisting of Extract key information, leverage text continuation, etc. is stated clearly and is interesting.

**Weaknesses:**

- The point behind Figure 6 is not clear. It'd be useful if authors can say which metric (e.g. Aesthetic, FID, etc.) is improved by CoT and provide more explanation about the performance of the other methods as well.
- Experimental results are hard to understand and results are not fully explained. I encourage the authors to remove the section on text to video results and focus more on the learnings from the text to image experiments.
- Studied methods are not clearly introduced e.g. baseline, t2t-blip, cot-d, cot, etc.
- Results reported in Table 2 are not discussed throughly. e.g. the metrics are not explained It's not clear why certain models perform superior/inferior for certain metrics. It's not clear when we expect a certain model to perform better than the others.
- Results reported in Table 4 are not explained clearly either. e.g. the delta between CLIP score of cot and cot-d looks within the margin of error. Not clear if one is performing superior than the other. Not clear how we should be thinking of the three metrics reported and their importance for various types of tasks.
- It would have been useful if authors would have clarified the contributions of their proposed method through concrete examples over other alternatives. If this requires additional space authors can add an appendix section to the paper.

**Questions:**

Please read the points in the weakness section.

**Details Of Ethics Concerns:**

No particular to this submission but for any text to image generation safety and privacy concern may apply. e.g. consider using text to image to generate violent or other types of sensitive/illegal content.

---

### Official Review · Reviewer_zdHw · 2023-11-11

**Soundness:** 2 fair
**Presentation:** 2 fair
**Contribution:** 2 fair
**Rating:** 3
**Confidence:** 3

**Summary:**

The paper investigates the relationship between the prompt details and the quality of synthetic images generated by a diffusion model. To achieve this, the authors propose refining the provided prompt using the Chain of Thought mechanism, which relies on Large Language Models (LLMs). They then proceed to demonstrate the effectiveness of this proposed method.

**Strengths:**

1. The paper introduces a method for refining the prompt, allowing for the addition of additional details to enhance the generation of images with greater complexity.

2. The experiments illustrate the need for further refinement of a given prompt by including more details.

**Weaknesses:**

1. The presentation is not clear and some details of the method are missing, e.g., the training details, what is the adapter introduced in the paper, and how to fine tune it? what are the boosters introduced in the Table?

2. What are the main benefits to further fine-tune a LLM on a specific task shown in the paper. Basically, the LLM, such as ChatGPT, is already able to do the text-continuation and text-revision. It might be better to include some comparison experiments to show the necessary to further fine-tune a LLM.

3. The proposed Chain of Thought mechanism is still based on the capability of LLMs, so how does it avoid the drawback mentioned by authors that pre-trained LLMs tend to generate extraneous or irrelevant information.

4. I am confused about the proposed relation between the quality of textual prompt and the quality of synthetic image, where the performance  of diffusion model is significantly contingent upon the quality of textual inputs. The main influence factor from the textual prompt is more about details of the description, while the details of a prompt cannot affect the performance of a pre-trained diffusion model. The text prompt shown in the Figure 1 is really confused.

**Questions:**

Please see above weaknesses.